# Laparoscopically assisted versus open oesophagectomy for patients with oesophageal cancer—the Randomised Oesophagectomy: Minimally Invasive or Open (ROMIO) study: protocol for a randomised controlled trial (RCT)

Rachel C Brierley,[1] Daisy Gaunt,[2] Chris Metcalfe,[2] Jane M Blazeby,[3,4] Natalie S Blencowe ![ORCID],[3,4] Marcus Jepson,[5] Richard G Berrisford,[6] Kerry N L Avery ![ORCID],[5] William Hollingworth,[7] Caoimhe T Rice,[7] Aida Moure-Fernandez,[7] Newton Wong,[8] Joanna Nicklin,[4] Anni Skilton,[9] Alex Boddy,[10] James P Byrne,[11] Tim Underwood,[11] Ravi Vohra,[12] James A Catton,[12] Kish Pursnani,[13] Rachel Melhado,[14] Bilal Alkhaffaf ![ORCID],[14] Richard Krysztopik,[15] Peter Lamb,[16] Lucy Culliford,[1] Chris Rogers,[1] Benjamin Howes,[4] Katy Chalmers,[7] Sian Cousins,[7] Jackie Elliott,[17] Jenny Donovan,[5] Rachael Heys,[1] Robin A Wickens,[1] Paul Wilkerson,[4] Andrew Hollowood,[4] Christopher Streets,[4] Dan Titcomb,[4] Martyn Lee Humphreys,[6] Tim Wheatley,[6] Grant Sanders,[6] Arun Ariyarathenam,[6] Jamie Kelly,[11] Fergus Noble,[11] Graeme Couper,[16] Richard J E Skipworth,[16] Chris Deans,[16] Sukhbir Ubhi,[10] Robert Williams,[10] David Bowrey,[10] David Exon,[10] Paul Turner,[13] Vinutha Daya Shetty,[13] Ram Chaparala,[14] Khurshid Akhtar,[14] Naheed Farooq,[14] Simon L Parsons,[12] Neil T Welch,[12] Rebecca J Houlihan,[4] Joanne Smith,[6] Rachel Schranz,[11] Nicola Rea,[16] Jill Cooke,[10] Alexandra Williams,[13] Carolyn Hindmarsh,[14] Sally Maitland,[12] Lucy Howie,[15] Christopher Paul Barham[4]

For numbered affiliations see end of article.

**Correspondence to**
Dr Rachel C Brierley;
r.brierley@bristol.ac.uk

## ABSTRACT

**Introduction** Surgery (oesophagectomy), with neoadjuvant chemo(radio)therapy, is the main curative treatment for patients with oesophageal cancer. Several surgical approaches can be used to remove an oesophageal tumour. The Ivor Lewis (two-phase procedure) is usually used in the UK. This can be performed as an open oesophagectomy (OO), a laparoscopically assisted oesophagectomy (LAO) or a totally minimally invasive oesophagectomy (TMIO). All three are performed in the National Health Service, with LAO and OO the most common. However, there is limited evidence about which surgical approach is best for patients in terms of survival and postoperative health-related quality of life.

**Methods and analysis** We will undertake a UK multicentre randomised controlled trial to compare LAO with OO in adult patients with oesophageal cancer. The primary outcome is patient-reported physical function at 3 and 6 weeks postoperatively and 3 months after randomisation. Secondary outcomes include: postoperative complications, survival, disease recurrence, other measures of quality of life, spirometry, success of patient blinding and quality assurance measures. A cost-effectiveness analysis will be performed comparing LAO with OO. We will embed a randomised substudy to

## Strengths and limitations of this study

► The Randomised Oesophagectomy: Minimally Invasive or Open (ROMIO) study will compare laparoscopically assisted oesophagectomy with open oesophagectomy, which are the procedures most relevant to UK practice.
► We will assess the quality of the surgery, using operative images and pathology.
► The primary outcome (recovery of physical function up to 3 months) considers what matters most to patients about having an oesophagectomy.
► Patients will be blinded to their surgical procedure for 6 days postoperatively (using large dressings) to achieve an unbiased assessment of pain, but it will not be possible to blind patients for the primary outcome.
► ROMIO incorporates a randomised substudy to collect data on totally minimally invasive oesophagectomy, which is an evolving technique.

evaluate the safety and evolution of the TMIO procedure and a qualitative recruitment intervention to optimise patient recruitment. We will analyse the primary outcome

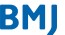

using a multi-level regression model. Patients will be monitored for up to 3 years after their surgery.

**Ethics and dissemination** This study received ethical approval from the South-West Franchay Research Ethics Committee. We will submit the results for publication in a peer-reviewed journal.

**Trial registration number** ISRCTN10386621.

## INTRODUCTION

In the UK, about 8900 people are diagnosed with oesophageal cancer each year and the incidence is increasing.[1] Surgical removal of the oesophagus (oesophagectomy), with or without chemo(radio)therapy, is currently the most commonly recommended treatment for patients whose cancer is confined to the oesophagus and the local lymph nodes and who are fit to undergo major surgery. The objective of treatment is a surgical cure but only about 40%–50% of patients survive for 3 years or more following treatment.[1] The surgical procedure depends on the location and size of the tumour and individual surgeon choice. There are a number of different surgical approaches used in the National Health Service (NHS), but the most commonly used procedure involves removing the bottom part of the oesophagus and part of the top of the stomach (the two-phase Ivor Lewis oesophagectomy). The remaining stomach is fashioned into a tube and brought up into the chest to replace the removed oesophagus.

In the past 10 years, there has been an increase in the use of minimally invasive surgical techniques and, according to the latest Association of Upper Gastro-Intestinal Surgeons audit, 42% of oesophagectomies were performed using laparoscopically assisted oesophagectomy (LAO) or totally minimally invasive oesophagectomy (TMIO).[2] However, it is uncertain whether laparoscopic surgery improves patient recovery after surgery or has any impact on survival.

Observational studies suggest that TMIO achieves the same survival benefit as open oesophagectomy (OO), but with better recovery and reduced rates of postoperative pneumonia,[3–5] although the apparent faster recovery may be due to the selection of fitter patients for the minimally invasive procedure. To date, seven randomised controlled trials (RCTs) comparing OO with LAO (n=2), or TMIO (n=4), or robot-assisted TMIO (n=1) have been conducted.[6–12] All had modest sample sizes (26–221 patients) and five out of the seven studies were conducted in a single centre (China=3, Austria=1, The Netherlands=1). The studies measured short-term primary outcomes such as pulmonary infection (n=2),[7 9] postoperative complications (n=4)[6 10–12] and duration of operation (n=1).[8] In one RCT, patients were randomised to a surgeon rather than procedure, meaning the treatment effect may be influenced by a difference in skill between surgeons choosing LAO and those choosing OO.[11] All but one of the RCTs[12] were at unclear risk of selection bias, either due to random sequence generation (n=3) or allocation concealment (n=6). The best evidence comes from two multicentre RCTs, MIRO (hybrid minimally invasive

versus open oesophagectomy for patients with oesophageal cancer) and TIME (traditional invasive vs minimally invasive oesophagectomy). MIRO randomised 207 participants in 12 French centres. Patients were randomised to OO (n=104) or LAO (n=103). They compared intraoperative and postoperative complications, classified as grade 2 or above using Clavien-Dindo, within 30 days. There was a lower incidence of complications in those allocated to LAO (36%) compared with those allocated to OO (64%, OR 0.31, 95% CI 0.18 to 0.55) However, patients were randomised using opaque envelopes in theatre after a preoperative laparoscopic investigation.[10] The TIME trial, conducted in five European centres, compared TMIO with OO in 115 patients[7] and reported a 70% reduction in pulmonary infection in the TMIO group in the first 2 weeks postoperatively (relative risk (RR) 0.30, 95% CI 0.12 to 0.76).[7] However, the TMIO procedure is not well-established in the UK.[13]

We are conducting a large, multicentre RCT (the ROMIO trial) to compare the clinical and cost-effectiveness of LAO versus OO. The trial will provide high-quality evidence, relevant to UK practice, of the risks and benefits of LAO, in terms of recovery, health-related quality of life (HRQoL), cost and survival. Incorporated into the study are:

▶ An assessment of the quality of the surgery performed, using intraoperative photos of the procedure and pathology reports.[14]
▶ An integrated qualitative QuinteT Recruitment Intervention to optimise recruitment.[15]
▶ A nested randomised substudy (designed as a phase 2b study within the innovation, development, exploration, assessment, and long-term study, IDEAL, framework) to investigate the safety and technical changes in TMIO.[16]

## METHODS AND ANALYSIS

We have used the Standard Protocol Items: Recommendations for Interventional Trials reporting guidelines in this protocol paper.[17]

### Study design

ROMIO is a multicentre RCT comparing OO with LAO in patients with oesophageal cancer. ROMIO will also include a nested randomised substudy in two centres to assess the efficacy of TMIO and review safety data, compared with OO and LAO. The substudy will also document how the technical aspects of TMIO evolve over time and whether the technique 'stabilises' over the course of ROMIO.

### Entry criteria

To ensure comparability between centres and surgeons, centres will only be included if they are undertaking at least 50 oesophagectomies per year and have a minimum of two surgeons participating in ROMIO. Surgeons will be assessed (by electronically submitting two unedited anonymised videos to the ROMIO study imaging team)

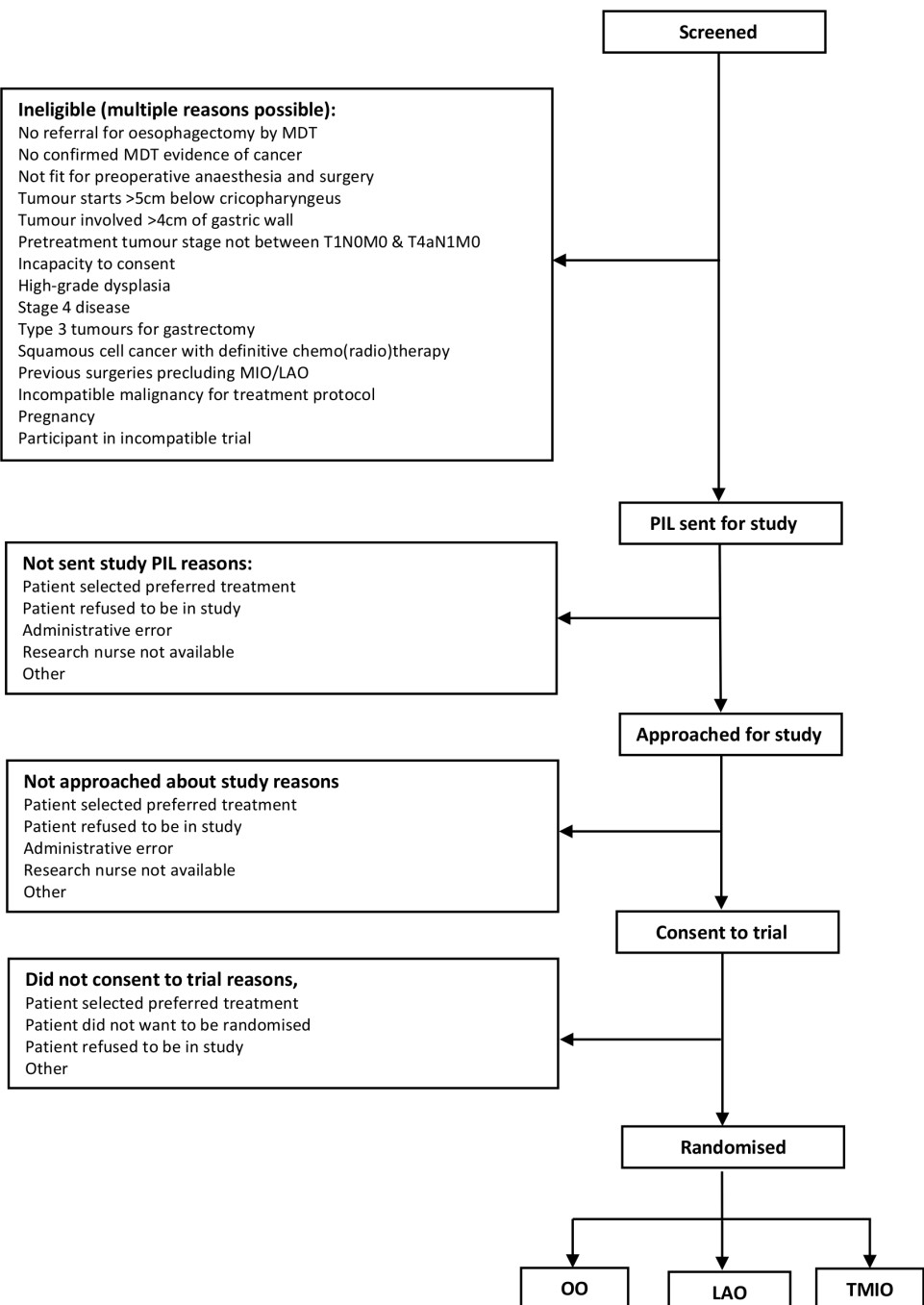

**Figure 1** Consort diagram outline. MDT, multidisciplinary team; PIL, patient information leaflet; TMIO, totally minimally invasive oesophagectomy; LAO, laparoscopically assisted oesophagectomy; OO, open oesophagectomy.

before they will be permitted to enrol their patients for ROMIO. Further details about this quality assurance (QA) measure has been described previously.[14]

### Inclusion criteria

We will screen all patients undergoing oesophagectomy (with or without neoadjuvant chemo(radio)therapy) in at least eight UK hospitals for eligibility (figure 1). We will include patients who are at least 18 years of age, with at least adenocarcinoma or squamous cell cancer of the oesophagus or oesophago-gastric junction, who have been referred for oesophagectomy by the multidisciplinary team after neoadjuvant chemotherapy or chemo(radio) therapy (any type). Patients will be included if their tumour is localised (has not spread beyond the local lymph nodes), is >5 cm below the cricopharyngeus (the muscle that keeps the oesophagus shut) and involves <4 cm of the stomach wall. Patients will only be included if they have been assessed as fit for surgery and are able to provide written informed consent.

### Exclusion criteria

Patients will be excluded if they have high-grade dysplasia or if the cancer has spread beyond the oesophagus (T4b)

or any stage with M1. All patients must be eligible for either open or minimally invasive surgery and must not be taking part in any other research that would interfere with the ROMIO protocol.

## Randomisation

The local research team will take written informed consent from participants. They will then randomise participants up to 2 weeks before their operation using a secure internet-based randomisation system. A computer programme will be used to generate the allocation sequence used for randomisation. Randomisation will be stratified by neoadjuvant treatment and site. Randomisation within blocks of varying size will prevent large imbalances in the number of patients in each treatment group. Participants will be randomised to receive either OO or LAO in a 1:1 ratio (with a varying block size of 6 or 8). In two centres, patients may also be randomised to receive TMIO, in a 1:1:1 ratio (with a varying block size of 6 or 9). The surgical team will be informed of the patient allocation after randomisation and before surgery (figure 1).

## Trial interventions

The intervention being compared in ROMIO is the surgical approach, that is, whether the surgeon uses large (OO) or smaller incisions (LAO or TMIO) to perform the operation (figure 2). Internally, the operation being performed is expected to be the same, regardless of the surgical approach used. Placement of a feeding jejunostomy or naso-jejunal tube will also at the surgeon's discretion, as well as the use of intra-abdominal and intrathoracic drains. Details of the surgical technique were established during the feasibility study and as part of the embedded QA study and are intended to be pragmatic.[14][18] OO will be performed using large incisions in both the abdomen and the chest (figure 2); the location and length of incisions are at each surgeon's discretion. LAO will be performed laparoscopically using 5 and/or 12 mm incisions (as many as needed, according to surgeon's preference) in the abdomen. One large incision will be made in the chest (figure 2). If a feeding jejunostomy tube is placed, this may be performed laparoscopically or by creating an abdominal incision (no bigger than 8 cm). In the two centres participating in the substudy, around 33% of patients will have a TMIO. In this approach, the surgeon will make small incisions in the abdomen and in the chest. For the abdominal part of the procedure, laparoscopic techniques will be used as described above. The surgeon will access the thoracic cavity using 12 and/ or 5 mm incisions (as many as needed) and perform the surgery thoracoscopically (figure 2).

Procedures to minimise diaphragmatic herniation (where one or more of the abdominal organs moves into the chest) can be performed at the surgeon's discretion. The anastomotic technique and methods to close the incisions are at the surgeon's discretion. Any deviations from the specified procedures must be fully documented and will be reviewed by the study management group.

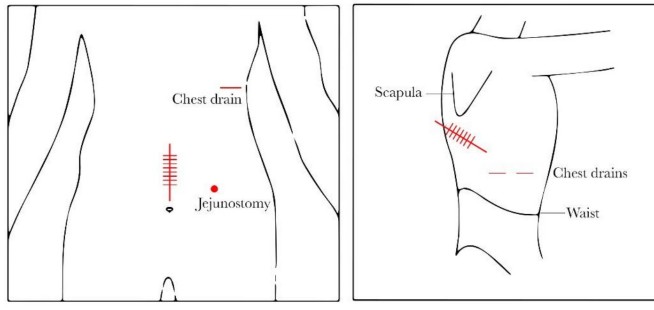

A. Open oesophagectomy (OO)

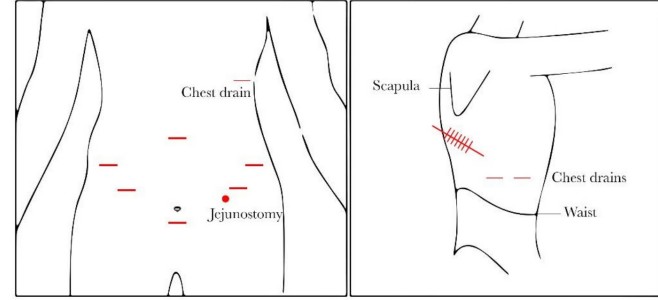

B. Laparoscopically assisted oesophagectomy (LAO)

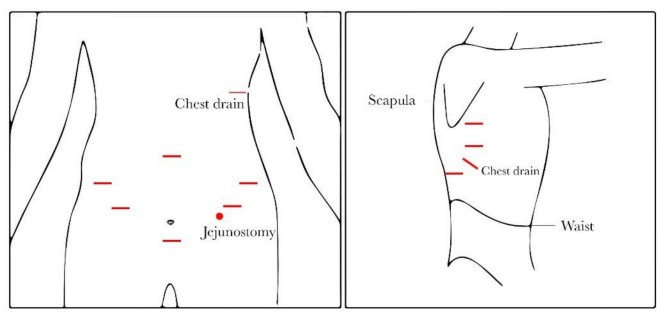

C. Totally minimally invasive oesophagectomy (TMIO)

**Figure 2** Diagrams representing the incisions the surgeon may make for the three different surgical approaches.

All surgical interventions will be carried out under general anaesthesia according to local hospital protocols. Patients will receive antibiotics and deep vein thrombosis prophylaxis according to local hospital policies. Co-interventions such as perioperative analgesia (eg, epidural anaesthesia or paravertebral catheters) and monitoring (eg, central or arterial lines) will be permitted according to the preferences of each centre.

Participants have the right to discontinue their part in the study at any time. In addition, the investigator may withdraw the participant from their allocated treatment group if, subsequent to randomisation, a clinical reason for not performing the surgical intervention is discovered. Participants withdrawn from their allocated intervention but willing to continue completing follow-up schedules will be encouraged to do so. All discontinuations and withdrawals will be documented.

## Primary outcome

The primary outcome is recovery of physical function assessed using the established, validated patient-reported European Organisation for Research and Treatment of

Cancer quality of life questionnaire (EORTC-QLQ-C30) at 3 and 6 weeks postsurgery and 3 months after randomisation.[19] This quality of life measure was selected as the key benefits of minimally invasive surgical techniques are the potential for less tissue damage and consequently less pain and a more rapid recovery of function.[20] Patient groups also indicated that quality of life is an outcome that is very important to them.

### Secondary outcome measures

Secondary outcomes will assess the efficacy of the OO and LAO in terms of morbidity, survival and safety. Secondary outcomes will include:

#### Survival
► Overall and disease-free survival for at least 2 years.

#### Complications
► All-cause short-term and long-term complications for up to 3 years after randomisation.[21]
► Any complications within 30 days of surgery, as assessed using the Clavien-Dindo System.[22]
► Length of hospital stay (defined as length of primary hospital stay plus readmission within 30 days/length of primary hospital stay plus length of hospital stay if discharged to community hospital).
► Forced expiratory volume in 1 s and forced vital capacity measured by spirometry, at baseline and days 3 and 6 postoperatively.

#### Cost-effectiveness
► Incremental net monetary benefit of LAO over OO 2 years after surgery.

#### Quality of life
► We will measure generic and disease-specific aspects of HRQoL using the following validated questionnaires at baseline, 6 days, 3 weeks and 6 weeks postsurgery and 3, 6, 9, 12, 18, 24 and (where possible) 36 months after randomisation:
  – EORTC QLQ-C30[19]—a questionnaire developed to assess the HRQoL of patients with cancer.
  – EORTC QLQ-OES18[23 24]—a questionnaire developed to assess the HRQoL for patients with oesophageal cancer.
  – Multidimensional fatigue inventory (MFI)-20[25 26]—a tool widely used to assess fatigue in patients with cancer.
  – EuroQOL EQ-5D-5L[27 28]—a widely used generic quality of life questionnaire.

We will also measure pain preoperatively and postoperatively at days 3 and 6 using the visual analogue scale (VAS).[29]

#### Quality assurance
We will assess QA of surgery (reported previously[14]) using the following:

► Intraoperative photographs will be taken at key stages throughout each procedure and submitted to the study database for ongoing monitoring of the operations. Anonymised images will be reviewed and rated by surgical assessors.
► Histopathological measures assessed by pathologists blinded to the treatment allocation, including length of the oesophagus; total counts of nodes—all and malignant (positive) nodes; details of resection margins and pT staging. The slides of 10% of all cases from each centre will be reviewed by the lead pathologist.
► Success of patient blinding during the first 6 days postoperatively, assessed using the Bang Blinding Index.[30]

Outcome data will be collected onto case report forms (unless questionnaires are specified) and entered onto a study-specific database for data cleaning and analysis.

Data about adverse events will be collected and reported in accordance with sponsor and regulatory requirements.

### Patient and public involvement

Patient groups were consulted at the design stage, the feedback from these patient consultations shaped the primary outcome and other aspects of the study. We have patient representatives as grant co-applicants and as independent members of the trial steering committee; in addition to this, we will regularly consult patient and public groups about different aspects of the study as it is ongoing. Patients have provided feedback on the burden involved in participating in the research.

### Methods used to minimise bias

Patients will be blinded to their treatment allocation by covering all potential incision sites for all surgical approaches (regardless of the actual operation performed) with large dressings for the first 6 days postoperatively. On day 6, patients will be asked to complete a booklet containing all of the quality of life questionnaires (QLQ-C30, OES-18, MFI-20, EQ-5D-5L) and a pain assessment using the VAS. Success of blinding will be assessed using the Bang Blinding Index.[30] Due to the nature of the study intervention, it is not possible to blind patients for the completion of the quality of life questionnaires at the primary outcome timepoints. However, patients have not had this surgery before so will not have anything to compare it to; furthermore, participants in the study are unlikely to have a strong preference for one approach over another.

Pathologists assessing QA of surgery will also be blinded to the randomised allocation. As the intervention is surgery, there may be variation in surgeon skill or surgical technique. This will be managed by stratifying the randomisation by centre. Surgical QA is described in more detail elsewhere.[14]

Loss to follow-up will be minimised by maintaining regular contact with patients (by telephone and post) to complete follow-up questionnaires. No additional visits are required for the study.

## Sample size

Two hundred three participants in each of the LAO and OO groups will allow a minimum clinically important difference of 0.4 SD on the primary outcome to be detected with >90% power at the 5% significance level, allowing for 15% of participants not following their allocated procedure, and 10% failing to complete the primary outcome. We anticipate that approximately 40 additional patients will be randomised to TMIO in the nested IDEAL phase IIb substudy to allow us to describe and evaluate changes in technique.

## Statistical analysis

The results will be reported according to the Consolidated Standards of Reporting Trials guidelines, including the extension for patient-reported outcomes.[31] We will analyse the data according to the intention-to-treat principle, in that the groups compared will be based on allocated treatment irrespective of the actual operation that the patient had. Participants missing all three assessments contributing to the primary outcome measure will not be included in the primary analysis, but we will use sensitivity analyses to investigate the potential impact of any missing data. We will adjust analyses for treatment centre, whether the participant underwent neoadjuvant chemo(radio)therapy and the baseline value of the outcome under comparison. We will prepare and make publicly available a detailed analysis plan prior to locking the database.

The primary outcome measure (difference between LAO and OO treatment groups) will be the reported difference in mean EORTC-QLQ-C30 scores for recovery of physical function (with 95% CI and p value). The difference in mean scores will be estimated as the coefficient of a binary variable distinguishing the two treatment groups, in a multilevel regression model, with covariates as detailed. This analysis will be conducted separately for data from the feasibility and main trials, and the two treatment effect estimates pooled as a weighted average. The same approach will be adapted to the assessment of pain during the 6 days postoperatively and other measures of HRQoL.

In addition, we will compare postoperative length of stay and accommodate the skewed distribution of this measure by a log-transformed analysis model presenting ratios of geometric means, 95% CI and p value.

We will present frequencies of the key treatment complications by treatment allocation. Severity of treatment complications will be compared between allocated treatment groups by scoring each patient according to their most severe Clavien-Dindo category and estimating the difference as an OR using ordered logistic regression.

We will use proportional hazards regression to estimate the treatment difference in overall and disease-free survival. A Kaplan-Meier plot will present survival over time in the OO and LAO groups.

## Subgroup analyses

A subgroup analysis will investigate whether the relative effects of OO and LAO differ according to whether a participant underwent neoadjuvant chemotherapy/chemo(radio)therapy beforehand.

## Analysis of the nested IDEAL phase IIB substudy

Data about the TMIO group will be collected and reported separately to the comparison between the OO and LAO groups, these patients will not be included in the main analysis. We will document the inclusion/exclusion of patients for the TMIO procedure and any reasons for not performing the TMIO operation according to the randomised allocation. We will document the complications of TMIO and perform some analyses of safety and adverse events compared with the OO and LAO groups. We will document the evolution of the technical aspects of this procedure according to the IDEAL phase IIb framework.

## Cost-effectiveness analysis

We will convert EQ-5D-5L[18 19] responses to utilities using the National Institute for Health and Care Excellence-recommended UK tariff at the time of analysis.[32] These will be combined with survival data to calculate quality-adjusted life years, adjusted for differences in baseline EQ-5D utility scores.[33] We will estimate theatre costs by collecting detailed information on equipment used and staff time (eg, surgeons, anaesthetist, scrub nurse). We will collect information on intensive care resource use and re-interventions during the initial hospital stay. We will also collect and analyse data on healthcare resources used in subsequent inpatient stays, outpatient visits, general practitioner visits and other community health services. We will use nationally available unit costs to value resource use where available.

We will perform the cost-effectiveness analysis on an intention-to-treat basis from both an NHS perspective, and a wider personal and social care perspective. We will estimate the cost-effectiveness of LAO compared with OO by calculating the incremental net monetary benefit and, if appropriate, the incremental cost-effectiveness ratio. We will present uncertainty in these estimates using a cost-effectiveness acceptability curve and/or cost-effectiveness ellipses. We will perform cost analyses at 3 and 24 months after randomisation. At 24 months, we will discount the cost estimates at the rate recommended by HM Treasury at the time of analysis.[34] We will conduct one-way sensitivity analyses including varying the discount rate. Where appropriate, we will use simple or multiple imputation techniques for missing data.

## ETHICS AND DISSEMINATION

All study interventions are already routinely used in the NHS. We will disseminate the findings by usual academic channels, that is, presentation at international meetings and peer-reviewed publications. We will write a full

report for the funder on the completion of the study and we will provide a lay summary of the results to patient organisations.

## Study progress

Recruitment started in October 2016 and we have recruited 277 patients, with an additional 32 TMIO patients (correct on 3 April 2019). We have agreed with the funder to continue recruitment until September 2019, beyond the planned completion of recruitment in November 2018. We will also include approximately 120 patients recruited to the feasibility study, having secured permission to continue recruitment to that study whilst the main trial was being set up.

## Major changes to the study protocol

Since the first study protocol was approved by the Research Ethics Committee (the current version is V.7.0, 25 October 2018), we have made the following changes:

► We updated the expected adverse events section to reflect the results of an international consensus paper on standardising reporting of complications of oesophagectomy[21] and to clarify that we will not collect events related to chemotherapy.

► We included information about plans to link with external registries (Intensive Care National Audit Research Centre, Public Health England, Information Services Division) to obtain more detailed data about ROMIO patient care in hospitals, for the purpose of economic analysis and to capture information on acute postoperative complications and recovery.

**Author affiliations**
[1]Clinical Trials and Evaluation Unit, Bristol Trials Centre, University of Bristol, University of Bristol, Bristol, UK
[2]Bristol Randomised Trials Collaboration, Bristol Trials Centre, University of Bristol, Bristol, UK
[3]Centre for Surgical Research, School of Social and Community Medicine, University of Bristol, Bristol, UK
[4]Division of Surgery, University Hospitals Bristol NHS Foundation Trust, Bristol, UK
[5]School of Social and Community Medicine, University of Bristol, Bristol, UK
[6]Upper GI Surgery, Derriford Hospital, Plymouth, UK
[7]Bristol Medical School: Population Health Sciences, University of Bristol, Bristol, UK
[8]Department of Cellular Pathology, North Bristol NHS Trust, Southmead Hospital, Bristol, UK
[9]Medical Illustration, University Hospitals Bristol NHS Foundation Trust, Bristol, UK
[10]Department of Surgery, Leicester Royal Infirmary, Leicester, UK
[11]Division of Surgery, University Hospital Southampton NHS Foundation Trust, Southampton, UK
[12]Department of General Surgery, Nottingham City Hospital, Nottingham, UK
[13]Department of Upper GI Surgery, Royal Preston Hospital, Preston, UK
[14]Department of Oesophago-Gastric Surgery, Salford Royal NHS Foundation Trust, Salford, UK
[15]Gastroenterology Department, Royal United Hospital Bath NHS Trust, Bath, UK
[16]General Surgery Department, Royal Infirmary of Edinburgh, Edinburgh, UK
[17]Gastro-Oesophageal Support and Help Group, Bristol, UK

**Acknowledgements** The authors would like to thank the large teams at each hospital who work on the ROMIO study. The hospitals involved with ROMIO are: University Hospitals Bristol NHS Foundation Trust, Bristol, England; University Hospitals Plymouth NHS Trust, Plymouth, England; University Hospital Southampton NHS Foundation Trust, Southampton, England; NHS Lothian, Edinburgh, Scotland; University Hospitals of Leicester NHS Trust, Leicester, England; Lancashire Teaching Hospitals NHS Foundation Trust, Preston, England; Salford Royal NHS Foundation Trust, Manchester, England; Nottingham University Hospitals NHS Trust, Nottingham, England; Royal United Hospitals Bath NHS Foundation Trust, Bath, England. The authors would also like to thank Maria Pufulete, Madeleine Clout and Kate Harris for commenting on the draft manuscript.

**Collaborators** The study is overseen by an independent steering committee and an independent data monitoring committee. Steering Committee membership: Craig Ramsay (Chair, Professor in Health Services Research, University of Aberdeen, Scotland); Tony Ingold (Trustee, Oesophageal Patients Association, England); Heike Grabsch (Professor of Gastrointestinal Pathology, Maastricht University, The Netherlands); William Allum (Consultant Upper Gastrointestinal Surgeon, Royal Marsden NHS Foundation Trust, London, England); Richard Hardwick (Consultant Upper Gastrointestinal Surgeon & Lead Clinician for Upper GI Cancer, Cambridge University Hospitals NHS FT, Cambridge, England); Colin Green (Professor of Health Economics, University of Exeter, Exeter, England); Helen Marshall (Principal Statistician, University of Leeds, Leeds, England). Data Monitoring Committee membership: Judith Bliss (Chair, Professor of Clinical Trials, Institute of Cancer Research, London, England); William Robb (Consultant Upper Gastrointestinal Surgeon, Beaumont Hospital & Royal College of Surgeons in Ireland, Dublin, Ireland); Derek Alderson (President of the Royal College of Surgeons and Emeritus Professor of Surgery, University of Birmingham, Birmingham, England).

**Contributors** All authors critically revised manuscript for important intellectual content and approved the manuscript. RCB: developed the study design and aims, drafted the protocol and initiated the study, drafted the manuscript and oversaw study conduct and acquisition of data. DMG: developed the study design and aims, drafted the protocol and initiated the study. DMG also oversaw statistical aspects and analyses for the study. NSB: developed the study concept and obtained grant funding, developed the study design and aims, drafted the protocol and initiated the study, oversaw the embedded quality assurance study. MJ: developed the study design and aims, drafted the protocol and initiated the study, oversaw the embedded qualitative study. KA: developed the study concept and obtained grant funding, developed the study design and aims, drafted the protocol and initiated the study, oversaw patient and public involvement. LC: developed the study design and aims, drafted the protocol and initiated the study. WH: developed and oversaw health economic aspects of the study. CR: developed the health economics. NW: developed the study concept and obtained grant funding, developed the study design and aims, drafted the protocol and initiated the study and oversaw pathology quality assurance in the study. JB: developed the study concept and obtained grant funding, developed the study design and aims, drafted the protocol and initiated the study, oversaw the embedded quality assurance study and clinical aspects of the study. JD: developed the study concept and obtained grant funding, developed the study design and aims, drafted the protocol and initiated the study and oversaw the embedded qualitative study. AM-F: developed the study design and aims, drafted the protocol and initiated the study and developed the health economics aspects of the study. JE developed the study concept and obtained grant funding and contributed to PPI aspects of the study. AS: developed the study design and aims, drafted the protocol and initiated the study and oversaw the embedded quality assurance study. RH, JN: developed the study design and aims, drafted the protocol and initiated the study and acquired data for the study. CM, RGB, CR, BH: developed the study concept and obtained grant funding, developed the study design and aims, drafted the protocol and initiated the study. CM: oversaw statistical aspects and analyses for the study. KC, SC: conducted the systematic review. RH, RW: conducted the study. JPB, TU, PL, AB, KP, RM, BA, RV, JC, RK: was a Principal Investigator and acquired data for the study. PW, AH, CS, DT, LH, TW, GS, AA, JK, FN, GC, RS, CD, SU, RDW, DB, DE, PT, VS, RC, KA, NF, SP, NW, JS, RS, NR, JC, AW, CH, SM, LH: acquired data for the study. CPB: developed the study concept and obtained grant funding, developed the study design and aims, drafted the protocol and initiated the study, oversaw clinical aspects of the study, oversaw the embedded quality assurance study and was Chief Investigator for the study.

**Funding** This research was funded by the National Institute for Health Research (NIHR) Health Technology Assessment (HTA) programme (project number 14/140/78). The authors also received funding support from the Bristol Biomedical Research Centre, the Medical Research Council (MRC) ConDuCT-II (Collaboration and innovation for Difficult and Complex randomised controlled Trials In Invasive procedures) Hub for Trials Methodology Research (MR/K025643/1) and the NIHR Biomedical Research Centre at University Hospitals Bristol NHS Foundation Trust and the University of Bristol. This study was designed and delivered in collaboration with the Clinical Trials and Evaluation Unit (CTEU) and the Bristol Randomised Trials

Collaboration (BRTC), both are UKCRC registered clinical trials units, which are known collectively as the Bristol Trials Centre (BTC), and receive National Institute for Health Research Clinical Trials Unit (CTU) support funding.

**Disclaimer** The views and opinions expressed therein are those of the authors and do not necessarily reflect those of the HTA programme, NIHR, NHS or the Department of Health. The funder and sponsors approve any amendments to the study but have no direct involvement in study design; collection, management, analysis and interpretation of data; writing of the report and the decision to submit this report for publication.

**Competing interests** None declared.

**Patient consent for publication** Not required.

**Ethics approval** This study received approval from South-West Frenchay Research Ethics Committee (REC, study ref: 184167).

**Provenance and peer review** Not commissioned; externally peer reviewed.

**ORCID iDs**
Natalie S Blencowe http://orcid.org/0000-0002-6111-2175
Kerry N L Avery http://orcid.org/0000-0001-5477-2418
Bilal Alkhaffaf http://orcid.org/0000-0001-5751-1846

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
