## [Reviewer comments · BMJ Open]

ARTICLE DETAILS

TITLE (PROVISIONAL)	Laparoscopically assisted vs open oesophagectomy for patients with oesophageal cancer – the ROMIO (Randomised Oesophagectomy: Minimally Invasive or Open) study: protocol for a randomized controlled trial (RCT)
AUTHORS	Brierley, Rachel; Gaunt, Daisy; Metcalfe, Chris; Blazeby, Jane; Blencowe, Natalie; Jepson, Marcus; Berrisford, Richard G.; Avery, Kerry; Hollingworth, William; Rice, Caoimhe; Moure-Fernandez, Aide; Wong, Newton; Nicklin, Joanna; Skilton, Anni; Boddy, Alex; Byrne, James; Underwood, Tim; Vohra, Ravi; Catton, James; Pursnani, Kish; Melhado, Rachel; Alkhaffaf, Bilal; Krysztopik, Richard; Lamb, Peter; Culliford, Lucy; Rogers, Chris; Howes, Ben; Chalmers, Katy; Cousins, Sian; Elliott, Jackie; Donovan, Jenny; Heys, Rachael; Wickens, Robin; wilkerson, paul; Hollowood, Andrew; streets, christopher; titcomb, dan; humphreys, lee; wheatley, tim; sanders, Grant; ariyarathenam, arun; kelly, jamie; noble, fergus; couper, graeme; skipworth, richard; deans, chris; ubhi, sukhbir; Williams, R; bowrey, david; exon, david; turner, paul; daya shetty, vinutha; chaparala, ram; akhtar, khurshid; farooq, naheed; Parsons, Simon; welch, neil; Houlihan, Rebecca; Smith, Jo; Schranz, Rachel; rea, nicola; cooke, jill; williams, alexandra; hindmarsh, Carolyn; maitland, sally; howie, lucy; Barham, C

VERSION 1 – REVIEW

REVIEWER	Alexander Phillips Northern Oesophagogastric Unit, Newcastle upon Tyne UK
REVIEW RETURNED	07-May-2019

GENERAL COMMENTS	This is a clinical trial evaluating an important question that is already well underway. It is clearly written and easy to follow. There are only some minor questions- who are the videos submitted too- regarding surgical technique? As per the evaluation of the study by NIHR- I think it would have been better not to included SCC and adenocarcinoma, and allowing the wide variation of surgical technique does detract from the overall applicability of the study. Whilst Clavien- Dindo is to be used for complication categorisation- it would be good to define what constitutes "major" at the outset. The MIRO trial chose C-D 2 and above (which conveniently led to a statistical difference) whilst most would state C-D 3 + is the definition of major.
--

REVIEWER	Jonathan Shenfine
-----------------	-------------------

	Flinders Medical Centre, Adelaide, Australia
REVIEW RETURNED	21-May-2019

GENERAL COMMENTS	This is a very well written protocol in terms of the study methodology. The technical aspects of the study and the QA associated with these are not presented but have been reported in other articles. I have no issues with the scientific quality of this work. My only concerns with the presented study is in terms of likely low numbers when analysing differences between histological subtypes, and the variety of neoadjuvant regimens allowed. Equally the sub-study of TMIO may not reach statistical significance due to numbers.
--

VERSION 1 – AUTHOR RESPONSE

Reviewer 1 Comments to Author:

This is a clinical trial evaluating an important question that is already well underway. It is clearly written and easy to follow.

Reply: We thank the reviewer for their support

There are only some minor questions- who are the videos submitted too- regarding surgical technique?

Reply: The videos are submitted electronically to the ROMIO study imaging team. All videos are then anonymised and reviewed by the Chief Investigator. Videos are also reviewed by a second assessor. More information about the assessment of videos has been published elsewhere (Blencowe, N.S., et al., Protocol for developing quality assurance measures to use in surgical trials: an example from the ROMIO study. *BMJ Open*, 2019. 9(3): p. e026209.)

Revisions: We have edited the “Entry criteria” paragraph on p9 as follows:

Surgeons will be assessed (by electronically submitting two unedited anonymised videos to the ROMIO study imaging team) before they will be permitted to enrol their patients for ROMIO. This Further details about this quality assurance (QA) measure has been described previously.

As per the evaluation of the study by NIHR- I think it would have been better not to included SCC and adenocarcinoma, and allowing the wide variation of surgical technique does detract from the overall applicability of the study.

Reply: We thank the reviewer for this comment. We wanted the pragmatic ROMIO trial to reflect current UK practice where oesophagectomy is the treatment of choice for oesophageal cancer, whether that is squamous cell carcinoma or adenocarcinoma. Both the TIME and MIRO trials included both squamous cell carcinoma and adenocarcinoma.

We have documented both the open and lap-assisted surgical procedures including mandatory, discretionary, and prohibited aspects for each. We are also collecting photographs of the key stages in each surgical procedure. We are confident that we will be able to present our results and relate these to a well specified and consistently achieved difference in surgical approach. There is likely to be variation between centres in other aspects of patient care, such as the nature of any enhanced

recovery programme. The potential for this variation to obscure any difference in outcome between open and lap-assisted surgery will be controlled, firstly by the stratification of the random allocation by centre, and secondly by including centre effects in the statistical model used to estimate that difference.

Whilst Clavien- Dindo is to be used for complication categorisation- it would be good to define what constitutes "major" at the outset. The MIRO trial chose C-D 2 and above (which conveniently led to a statistical difference) whilst most would state C-D 3 + is the definition of major.

Reply: We agree with the reviewer's comment, however we are not intending to define "major" complications as was done in the MIRO trial. Instead, we intend to present the statistics for each of the Clavien-Dindo categories (Normal recovery; Grade I / Grade II; Grade IIIa / Grade IIIb; Grade IVa / Grade IVb; Grade V) and discuss the results accordingly.

Reviewer 2 Comments to Author:

This is a very well written protocol in terms of the study methodology. The technical aspects of the study and the QA associated with these are not presented but have been reported in other articles. I have no issues with the scientific quality of this work.

Reply: We thank the reviewer for their support

My only concerns with the presented study is in terms of likely low numbers when analysing differences between histological subtypes, and the variety of neoadjuvant regimens allowed.

Reply: We have only one planned sub-group analysis, comparing any intervention effect between those who did and those who did not undergo neoadjuvant treatment; we agree that the study is not large enough to detect differences between the variety of neoadjuvant treatments in the relative effects of the open and hybrid procedures. We are not anticipating and have no plans to investigate whether the relative effects of the open and hybrid procedures on post-surgical recovery will differ by histological subtype, and again we agree that our study, despite having a larger sample size than other RCTs in the area, will not have sufficient statistical power to detect such sub-group effects.

Equally the sub-study of TMIO may not reach statistical significance due to numbers.

Reply: The sub-study of TMIO is aimed at providing early descriptive data on this procedure in an unselected cohort of patients, including an examination of the stability of the procedure methodology and the safety and was therefore never intended to reach statistical significance. These patients will not be included in the main analysis. We apologise for the confusion and have made revisions for clarification.

Revisions: We have edited the text in the following paragraphs:

"Sample size" (p15)

We anticipate that approximately 40 additional patients will be randomised to TMIO in the nested IDEAL 2b sub-study to allow us to describe and evaluate changes in technique.

"Analysis of the nested IDEAL 2b study" (p16)

Data about the TMIO group will be collected and reported separately to the comparison between the OO and LAO groups, these patients will not be included in the main analysis.

VERSION 2 – REVIEW

REVIEWER	Alexander Phillips
----------	--------------------

	Northern Oesophagogastric Unit Royal Victoria Infirmary Newcastle upon Tyne UK
REVIEW RETURNED	23-Jun-2019

GENERAL COMMENTS	Many thanks for addressing the comments
---